

# Insight into winter haze formation mechanisms based on aerosol hygroscopicity and effective density measurements

Yuanyuan Xie, Xingnan Ye*, Zhen Ma, Ye Tao, Ruyu Wang, Ci Zhang, Xin Yang, Jianmin Chen, Hong

Chen

Shanghai Key Laboratory of Atmospheric Particle Pollution and Prevention (LAP³), Department of

Environmental Science and Engineering, Fudan University, Shanghai 200433, China.

*Correspondence to: Xingnan Ye (yexingnan@fudan.edu.cn), Jianmin Chen (jmchen@fudan.edu.cn).

**Abstract**: We characterize a representative haze event from a series of periodic particulate matter (PM) episodes that occurred in Shanghai during winter 2014. Particle size distribution, hygroscopicity, and effective density were measured online, along with analysis of water-soluble inorganic ions and single particle mass spectrometry. Regardless of pollution level, the mass ratio of SNA/$PM_{1.0}$ (sulfate, nitrate, and ammonium) slightly fluctuated around 0.28 over the whole observation, suggesting that both secondary inorganic compounds and carbonaceous aerosols (including soot and organic matter) contributed substantially to the haze formation. Nitrate was the most abundant ionic species during hazy periods, indicating that $NO_x$ contributed more to haze formation in Shanghai than did $SO_2$. The calculated PM concentration from particle size distribution displayed a variation pattern similar to that of measured





$PM_{1.0}$ during the representative PM episode, indicating that enhanced pollution level was attributable to
the elevated number of larger particles. The number fraction of the near-hydrophobic group increased as
the PM episode developed, indicating accumulation of local emissions. Three "banana-shape" particle
evolutions were consistent with the rapid increase in $PM_{1.0}$ mass loading, indicating rapid size growth by
condensation of condensable materials was responsible for the severe haze formation. Both
hygroscopicity and effective density of the particles increased considerably with growing particle size
during the banana-shaped evolutions, indicating that secondary transformation of $NO_x$ and $SO_2$ was a
major contributor to the particle growth. Our results suggest that the accumulation of gas-phase and
particulate pollutants under stagnant meteorological conditions and subsequent rapid particle growth by
secondary processes, were primarily responsible for the haze pollution in Shanghai during wintertime.
**Keywords**: air pollution; size distribution; hygroscopic growth; secondary process; Shanghai.

## 30    1. Introduction

Atmospheric aerosol has significant influences on radiation balance and climate forcing of the
atmosphere (Wang et al., 2011;Wang et al., 2014c;Wu et al., 2016a;IPCC, 2013); as well as strong impacts
on visibility (Yang et al., 2012;Lin et al., 2014;Xiao et al., 2014) and public health (Heal et al., 2012) in
heavily polluted areas. Recent studies found that short-term exposure to haze pollution could cause airway
inflammation and aggravate respiratory symptoms in chronic obstructive pulmonary disease patients (Wu



et al., 2016b;Guan et al., 2016).
With the huge achievements in economic development and rapid urbanization over the past 30 years,
particulate pollution has become a major environmental concern in China. The most severe haze event
that occurred in the first quarter of 2013, spread over 1.6 million $km^2$ (Wang et al., 2014a). This event
motivated the release of the Action Plan on Prevention and Control of Air Pollution with the goal of
reducing $PM_{2.5}$ (particulate matter smaller than 2.5 μm in aerodynamic diameter) concentration by 15−25%
in        2017        against        2012        in        three        major        city        clusters
(http://english.mep.gov.cn/News_service/infocus/201309/t20130924_260707.htm). In order to reduce
the $PM_{2.5}$ concentration, extensive studies have been conducted to investigate the sources and formation
mechanisms of haze pollution in recent years (Ye et al., 2011;Sun et al., 2016;Qiao et al., 2016;Hu et al.,
2016;Li et al., 2016;Guo et al., 2014;Zheng et al., 2015;Guo et al., 2013;Wang et al., 2016;Peng et al.,
2016). However, the haze formation mechanisms and source appointment of fine particles remain
uncertain.
Guo et al. (2013) summarized historical reports from 2000 to 2008 in Beijing and found that the
origins of urban fine particles varied in different seasons: the contribution of primary emissions is
comparable to that of secondary formation during winter heating periods whereas secondarily produced
aerosols dominate the fine PM sources in other seasons. As an important type of primary emissions in
urban area, black carbon (BC) is primarily from incomplete fossil fuel combustion. Light absorption of



BC aerosols is increased after atmospheric aging by coating with secondary materials and restructuring
(Khalizov et al., 2009). Due to cooling effect at the surface and warming effect aloft, the enhanced light
absorption and scattering by aged BC particles stabilize the atmosphere, hindering vertical transport of
gaseous and particulate pollutants (Wang et al., 2013). BC aging occurs much more efficiently in the
presence of highly elevated gaseous aerosol precursors so that light absorption increases by a factor of
2.4 within 4.6 h under highly polluted conditions in Beijing, significantly exacerbating pollution
accumulation and strongly contributing to severe haze formation (Peng et al., 2016).
Due to the implement of several effective regulatory policies, the increasing trend of primary
emissions has been under control since the 11[th] five-year period. A growing number of studies suggested
that secondary production was the major contributor to the haze episodes in recent years (Shi et al.,
2014;Zhao et al., 2013;Zhang et al., 2015a;Huang et al., 2014), in contrast that primary emissions were
of great importance in some haze episodes (Niu et al., 2016). Guo et al. (2014) reported that the
development of PM episodes in Beijing was characterized by efficient nucleation and continuous particle
growth over an extend period dominated by local secondary formation. They attributed the continuous
growth of particle size and constant accumulation of particle mass concentration to the highly elevated
concentrations of gaseous precursors such as $NO_x$, $SO_2$, and volatile organic compounds (VOCs), while
the contribution from primary emissions and regional transport was negligible. However, the role of
regional transport of $PM_{2.5}$ in haze formation remains controversial (Li et al., 2015;Zhang et al., 2015b).





The most important advances in the understanding of urban PM formation were reviewed by Zhang
et al. (2015c). The concentrations of $SO_2$, $NO_x$, and anthropogenic source VOCs in Beijing and other
cities of the developing world are significantly higher than those in the urban areas of developed countries,
resulting in large secondary production of sulfate, nitrate, and SOA. Synergetic effects among various
organic and inorganic compounds may exist under highly polluted conditions, indicating different PM
formation rates between developing and developed urban regions. Indeed, a large enhancement of
particulate sulfate was typically observed during regional haze events in China (Chen et al., 2016;Wang
et al., 2015;Fu et al., 2008;Xie et al., 2015). Currently, the highly elevated sulfate concentration during
haze episodes cannot be fully explained by model simulations (Wang et al., 2014b;Chen et al., 2016).
Recently, a significant breakthrough made by Wang et al. (2016) has provided a reasonable explanation
in the high level of sulfate during haze episodes. It is revealed by their laboratory experiments that aqueous
oxidation of $SO_2$ by $NO_2$ proceeds more efficiently with the increase of $NO_2$ concentration whereas the
reaction is suppressed in acid conditions, because acid effect reduces the solubility of $SO_2$ and reaction
rate. The enhanced sulfate formation during severe haze periods in Beijing was attributable to aqueous
oxidation of $SO_2$ by $NO_2$ on hygroscopic fine particles under conditions of elevated RH and the
concentrations of $NH_3$ and $NO_2$, which was confirmed by the comparable $SO_2$ uptake coefficients for
sulfate formation from field and laboratory results.
The hygroscopic properties of ambient particles vary significantly depending on the origin of the air



masses and the atmospheric aging process. In urban air, the population of near-hydrophobic particles can
be assumed to consist largely of freshly emitted combustion particles containing high mass fractions of
soot and water-insoluble organic compounds (Swietlicki et al., 2008;Massling et al., 2009). In contrast,
secondary sulfate or nitrate aged particles are more-hygroscopic, and their relative abundance is primarily
responsible for the hygroscopic growth of ambient particles at elevated RH (Topping et al.,
2005;Aggarwal et al., 2007;Gysel et al., 2007). Thus, hygroscopicity can serve as a tracer of source origins,
mixing state, and aging mechanisms of ambient particles. For example, temporal variations of aerosol
hygroscopicity have helped the explanation of haze formation mechanisms in Beijing and Shanghai (Ye
et al., 2011;Guo et al., 2014).

Density is one of the most important physicochemical properties for atmospheric aerosols. Effective

density has served as a tracer for new particle formation and for the aging process in previous studies (Yin
et al., 2015;Guo et al., 2014). The ambient particles in urban areas are mostly complex mixtures of
elemental carbon (EC), organics (OC), and secondary inorganic aerosols (SIA) (Hu et al., 2012). The
effective densities of traffic particles are below 1.0 g cm$^{-3}$, and density decreases with the increase of
particle size because there are more voids between primary particles in relatively larger aggregates
(Momenimovahed and Olfert, 2015). The density of OC is in between those of EC and SIA, and varies
with source. The effective density of combustion particles increases by filling the voids in the agglomerate
particles with condensed semi-volatile materials, or by restructuring agglomerates with hygroscopic SIA



(Momenimovahed and Olfert, 2015;Zhang et al., 2008).
In this study, a combined HTDMA-APM system was used to investigate the variations of
hygroscopicity and effective density of submicrometer aerosols during winter 2014 in urban Shanghai. In
addition, cascade samples were collected and a single particle mass spectrometry was used to better
understand the hygroscopicity and density variations. The primary objectives of this study were to
investigate the particle growth mechanisms and to identify the contribution of local emissions during the
winter haze episode.

**2. Experimental**
**2.1. Sampling site**
The measurements were conducted from December 21, 2014 to January 13, 2015 at the Department of
Environmental Science and Engineering in the main campus of Fudan University (31.30°N, 121.5°E), a
representative urban site close to a sub-center of Shanghai (Ye et al., 2010). At a supersite about 100 m
away, $PM_{1.0}$ was monitored using a Thermo Scientific™ 5030 SHARP monitor. Trace gas pollutants were
monitored using Thermo Scientific™ i-series gas analyzers (43i for $SO_2$, 49i for $O_3$, 42i for $NO/NO_2/NO_x$),
and meteorological data were monitored using an automatic meteorological station (Model CAWS600,
Huayun Inc., China) (Yin et al., 2015). The concentrations of $PM_{2.5}$, $PM_{10}$, and CO were released by the
Shanghai Environmental Monitoring Center. The height of the Planet Boundary Layer (PBL) was





computed online using the NCEP Global Data Assimilation System (GDAS) model
(http://ready.arl.noaa.gov/READYamet.php).
**2.2. HTDMA-APM system**
Particle size distribution, hygroscopic growth factor (GF), and effective density were measured using
a custom-built HTDMA-APM system (Figure 1). The custom-built HTDMA (Hygroscopic Tandem
Differential Mobility Analyzers) mainly consist of two long DMAs (3081L, TSI Inc.), a humidifier (PD-
50T-12MSS, Perma Pure Inc.) and a Condensation Particle Counter (CPC, Model 3771, TSI Inc.). A
detailed description of the HTDMA is available in Ye et al. (2009). In this observation, particle number
size distribution in the range of 14−600 nm and hygroscopic growth at 83% RH for particles with dry
diameters of 40, 100, 220, 300, 350, and 400 nm were determined by HTDMA (in turn). The
determination of effective density by DMA-APM was described previously (Yin et al., 2015;Pagels et al.,
2009). Briefly, a combined system consisting of a compact Aerosol Particle Mass Analyzer (APM, Model
3601, Kanomax Inc.) and a CPC (Model 3775, TSI Inc.) was connected to the sample tubing through a
3-way electrical switch behind the upstream DMA (DMA1). The APM comprises two coaxial cylindrical
electrodes rotating at the same angular velocity. Charged aerosol particles of a certain diameter sized by
DMA1 are axially fed into the annular gap between the electrodes and experienced an outward centrifugal
force from the particle rotating and an inward electrostatic force from the high-voltage field between the
electrodes. Particles pass through the APM and are sent to the CPC when the two forces are balanced.



The mass of particles that pass through the APM is determined by the rotation rate and the applied voltage.
Effective densities for dry diameters of 40, 100, 220, and 300 nm were determined by the method of
DMA-APM in this study. The HTDMA-APM was operated alternatively in HTDMA mode and then
DMA-APM mode, for every 40 min.

Before the field observation, the HTDMA-APM was calibrated using 40–450 nm NIST-Traceable PSL

particles and ammonium sulfate. The measured HTDMA data were inversed with the TDMA$_{inv}$ algorithm
to obtain the actual GF distribution. This is   because the raw data are only a skewed and smoothed
integral transform of the actual growth factor probability density function (GF-PDF) (Gysel et al., 2009).
The hygroscopicity parameter κ was derived from the GF data after inversion with the TDMA$_{inv}$ algorithm
according to the κ-Köhler theory (Petters and Kreidenweis, 2007).
**2.3. SPAMS**

A Single Particle Aerosol Mass Spectrometry (SPAMS, Hexin Analytical Instrument Co., Ltd., China)

installed in the same room with the HTDMA-APM system was used to obtain the chemical and size
information of individual particles in the range of 0.2-2 μm. Detailed information on SPAMS is available
in Li et al. (2011). Briefly, ambient particles are drawn into a vacuum chamber through an aerodynamic
focusing lens and accelerated to a size-dependent terminal velocity. Sized particles are desorbed and
ionized by the pulsed desorption/ionization laser (Q-switched Nd: YAG, λ=266 nm) at the ion source
region. Both positive and negative mass spectra for a single particle are recorded by a bipolar time-of-



flight spectrometer. The single particle information was imported into YAADA (version 2.11,
www.yaada.org). Based on the similarities of the mass-to-charge ratio and peak intensity, particles were
classified using the ART-2a method.
**2.4. Ion chromatography**
Cascade aerosol samples for offline analysis were collected at the roof platform of the Environmental
Building using a 10-stage MOUDI sampler (Micro-Orifice Uniform Deposit Impactor, Model 110-NR,
MSP Corp., USA). Detailed description of the sampling, pretreatment, chemical analysis, and quality
control of this system is available in Tao et al. (2016). Briefly, cascade samples were collected every 24
h using the PALL7204 quartz filter as the collection substrate. Each filter was weighted with a BP211D
electronic balance at $25\pm1°C$ and $40\pm2\%RH$. The water extract of each sample was analyzed using an Ion
Chromatograph (Metrohm 883 basic IC plus, Switzerland) equipped with a third-party column heater
(CT-100, Agela Corp., China). Seven anions ($F^-$, $Cl^-$, $NO_2^-$, $Br^-$, $NO_3^-$, $SO_4^{2-}$ and $PO_4^{3-}$) were resolved
using a Metrosep A Supp 5-250/4.0 column at 35°C with an eluent of 3.2 mmol $L^{-1}$ $Na_2CO_3$ + 1.0 mmol
$L^{-1}$ $NaHCO_3$. Six cations ($Li^+$, $Na^+$, $NH_4^+$, $K^+$, $Ca^{2+}$, and $Mg^{2+}$) were separated by a Metrosep C4-250/4.0
column at 30°C with an eluent of 1.7 mmol $L^{-1}$ $HNO_3$ + 0.7 mmol $L^{-1}$ 2,6-pyridine dicarboxylic acid.

**3. Results and discussion**
**3.1. Periodic cycle of PM episodes during the observation period**



Figure 2 shows the temporal variations of PM mass loading during the winter observation (December
21, 2014 to January 13, 2015). The official data of $PM_{2.5}$ and $PM_{10}$ were blank on some clean days.
Meteorologically, our measurement was deployed in a typical winter period. The average concentrations
of $PM_{1.0}$, $PM_{2.5}$, and $PM_{10}$ were 57.3±37.0, 87.2±67.2, and 127.8±77.7 µg m$^{-3}$, respectively. About 62%
of hourly averaged $PM_{2.5}$ concentrations exceeded 75 µg m$^{-3}$ of the Chinese Grade II guideline (GB 3095-
2012), indicating heavy particle pollution in Shanghai during wintertime. The PM episodes exhibited a
clear periodic cycle of ~5 days. A similar feature was previously observed in Beijing (Guo et al., 2014).
At the beginning of each cycle, the $PM_{1.0}$ level was below 35 µg m$^{-3}$. During the clean period, the
differences among the concentrations of $PM_{1.0}$, $PM_{2.5}$, and $PM_{10}$ were insignificant. Occasionally the
measured $PM_{2.5}$ concentrations were larger than those of $PM_{10}$, possibly due to system error. However,
the particle mass concentration began to increase in the next few days, with $PM_{1.0}$ and $PM_{2.5}$ peaking at
over 100 and 200 µg m$^{-3}$, respectively. During the late episodes, the PM mass loading abruptly dropped,
due to change in the atmospheric dilution or wet deposition.
**3.2 Contributions of secondary inorganic aerosols to $PM_{1.0}$ mass loading**
Figure 3 illustrates the daily concentrations of sulfate, nitrate, and ammonium as a function of $PM_{1.0}$
mass loading. In general, the sum of concentrations of sulfate, nitrate, and ammonium (SNA) increased
linearly as $PM_{1.0}$ mass loading increased. It is noticeable that the $SNA/PM_{1.0}$ ratio slightly fluctuated
around 0.28, regardless of the pollution level. Because soil dust and sea salt made a negligible contribution



to the fine particle mass concentration in this study, the almost constant ratio of $SNA/PM_{1.0}$ indicates that
SNA and carbonaceous aerosols (including soot and organic matter) synchronously increased during haze
episodes. As the $PM_{1.0}$ concentration increased, the concentration of nitrate increased more rapidly than
sulfate so that it became the most abundant ionic species at $PM_{1.0} > 40$ µg m$^{-3}$. This indicated that $NO_x$
contributed more to haze formation in Shanghai than did coal-fired sources. Generally, the visibility
decreased with the increase in PM concentration, indicating photochemical activity began to weaken as
the development of haze episodes. The large increase in nitrate concentration may be attributable to
heterogeneous reaction on the preexisting particles. Nitrate formation is highly dependent on the surface
area of preexisting particles and is favored under $NH_3$-rich conditions (Chu et al., 2016). In contrast, Han
et al. (2016) reported that the mass ratio of nitrate to sulfate decreased with increase of $PM_{2.5}$ level and
that the sources of sulfate contributed more to haze formation in Beijing than mobile sources. This finding
suggests that the haze formation mechanism is likely different in Shanghai and Beijing. VOCs and $NO_x$
are exclusively from local emissions whereas regional transport is a big source of $SO_2$ under stagnant
atmosphere, due to their different atmospheric lifetimes (Guo et al., 2014). Considering the relatively
smaller contribution of sulfate, our results reveal that the accumulation and secondary transformation of
local emissions likely played a dominant role in this haze formation.
**3.3 Aerosol hygroscopicity and effective density during the observation period**
Figure 4a displays a box chart of the median hygroscopicity of each hygroscopic growth factor



distribution for different sizes. Considering all of the growth factor distributions collectively, the
hygroscopicity parameter κ increased with increase of the dry diameter, with an average κ of 0.161 at 40
nm and 0.338 at 300 nm. Assuming a two-component system of a model salt (ammonium sulfate, $\kappa_m =$
0.53) and an insoluble species (κ = 0), the volume fraction of hygroscopic species (ε) can be obtained
based on the Zdanovsldi-Stokes-Robinson (ZSR) mixing rule. The average ε was 0.3 for 40 nm particles,
suggesting that the primary particles or initial growth of freshly generated particles were dominated by
non-hygroscopic species. In contrast, the 300 nm particles were extremely aged, with more-hygroscopic
species.
Generally, the HTDMA-measured hygroscopicity was limited to the size range below 250 nm, and it
is common that the GF increases with increase of particle size. The increase of aerosol hygroscopicity
with size was attributed to the addition of more-hygroscopic SNA (Swietlicki et al., 2008;Ye et al., 2010).
Gasparini et al. (2006) reported that the GF first increased and then decreased with increase of particle
size, peaking at the diameter of 300 nm. In this study, the determination size range was extended to 400
nm and no decrease in GF was observed. We attribute the different hygroscopicity to the large emissions
of $SO_2$ and $NO_x$ in China, which were responsible for the strong formation of sulfate and nitrate. The
variation of hygroscopicity parameter κ was much greater for 40 nm particles. The particle population
with κ < 0.1 was attributed to fresh traffic particles (Ye et al., 2013). The considerable percentile of κ <
0.1 indicated that the 40 nm particle population was sometimes dominated by near-hydrophobic particles.



Figure 4b displays a box chart of median effective density for different particle sizes. The median
effective density varied in the narrow range of $\rho_{eff}$ = 1.35−1.41 g cm$^{-3}$ for 40−300 nm particle population.
The size distribution of particle density varied in the literature. Hu et al. (2012) and Yin et al. (2015)
reported that particle density increased as particle size increased while a contradictory trend was observed
by Geller et al. (2006) and Spencer et al. (2007). The difference was attributable to the contribution of
fresh traffic particles. Although the dominant accumulation mode particles have an effective density
greater than Aitken mode ones, the presence of a lower effective density group associated with emissions
from traffic exhaust might decrease the mean effective density to a value lower than that of Aitken mode
particles (Levy et al., 2014). Yin et al. (2015) reported that a quasi-monodisperse density distribution was
dominant for accumulation mode particles. In contrast, externally mixed aerosols with a lower density
group ($\rho_{eff}$ = ~1.0 g cm$^{-3}$) were often present in this observation, and were responsible for the decrease of
the mean effective density. The lower effective density group was attributed to fresh or slightly aged
traffic-related particles, because the number fraction of the lower density group increased as the
concentration of NO increased.
**3.4 Characteristics of a representative PM episode**
The PM episode from January 7 to 12 was a representative case of severe haze formation and
elimination processes. It can be divided into clean (January 7), transition (January 8), haze (January 9−11),
and post-haze (January 12) periods. During the transition from the clean to haze period (January 7 to 8),



both $PM_{1.0}$ and $PM_{2.5}$ concentrations increased slightly, with an average $PM_{1.0}/PM_{2.5}$ ratio of 0.65. A sharp
increase in $PM_{2.5}$ (of 125 µg m$^{-3}$) was observed from 6:00 to 12:00 local time on the morning of January
9. During the haze period, the concentration of $PM_{2.5}$ exceeded 115 µg m$^{-3}$ (medially polluted level,
HJ633-2012) for 63 h. On January 11, the hourly $PM_{2.5}$ concentration exceeded 250 µg m$^{-3}$, corresponding
to the severely polluted level.

Figure 5 displays the temporal evolution of particle size distribution in comparison with the measured

$PM_{1.0}$ concentration during the representative PM episode. The calculated PM concentrations ($PM_{cal}$)
were obtained based on the particle size distribution and average effective density of 1.39 g m$^{-3}$ in the
range of 14−600 nm measured in this study. In contrast to the fact that particle size distribution was
dominated by nanoparticles during the clean period, the burst of Aitken mode particles and subsequent
continuous growth to approximately 200 nm in diameter was observed three times during the haze period,
indicating that the presence of numerous larger particles is likely responsible for the severe particle
pollution (Guo et al., 2014). The importance of larger particles in haze formation is also illustrated by the
contour plot of the particle volume size distribution. The difference of total number concentration between
transition and haze periods was insignificant, whereas the volume concentration increased rapidly during
the haze period. This feature clearly demonstrates that the haze formation was closely correlated with
particle growth and elevated number of larger particles.

Interestingly, the particle mass concentration was sensitive to variations of wind speed and planetary





boundary layer (PBL). During the transition and haze periods, the wind speed decreased considerably
with insignificant change in prevailing wind (Figure S1). This indicated that outside transportation
became less and less significant. It is noteworthy that the temporal evolution of the particle mass
concentration was inversely correlated with the PBL height. The decreasing PBL provided a stagnant
atmosphere that favored the accumulation of local emissions. This finding reveals that the severe haze
pollution was likely triggered by the adverse meteorological conditions. The impact of decreasing PBL
height on haze formation can also be evidenced by the variations of trace gaseous species (Figure S2).
During the PM episode, the concentrations of $NO_2$, $SO_2$, and CO displayed variation trends similar to that
of the particle concentration. The fluctuations of trace gas concentrations were caused by primary
emission and secondary processes. Noticeably, the concentration of NO increased dramatically in rush
hours during the haze period, whereas it fluctuated slightly during the clean period; indicating that local
emissions were easily accumulated under stagnant atmosphere. In addition, the maximum concentration
of $O_3$ remained considerably higher during daytime, whereas it decreased significantly at night. The most
plausible explanation is that $O_3$ was consumed rapidly by the accumulating trace gases, such as $NO_x$, and
VOCs.
**3.5 Variations of hygroscopicity and effective density during the PM episode**
Figure 6 shows the averaged hygroscopicity and effective density for different pollution periods of the
PM episode. Regardless of the pollution period, near- hydrophobic particles were externally mixed with





some hygroscopic particles. During the clean period, the more-hygroscopic particles dominated the 40
nm particle population, indicating that the near-hydrophobic primary particles were rapidly dispersed due
to atmospheric dilution. The number fraction of the near-hydrophobic group for different sizes increased
as the PM episode developed, indicative of the increasing accumulation of local emissions. Notably, the
increase of the near-hydrophobic particles with the evolution of the PM episode become less significant
as particle size increased, indicating that primary emission exerted a more significant impact on smaller
particles than on larger ones. The median diameter of nascent traffic particles from various gasoline
sources ranged between 55 and 73 nm with an average of 65 nm (Momenimovahed and Olfert, 2015).
Therefore, the number fraction of the near-hydrophobic particles larger than 200 nm is not sensitive to
the accumulation of traffic emissions.

Interestingly, the variations of particle effective density for different sizes are in good agreement with

the hygroscopicity. The dominant peak of effective density distribution appeared at $\rho_{eff} = \sim1.5$ g cm$^{-3}$ for
40 nm particles in the clean period, indicating that they are highly aged with hygroscopic inorganic salts
(Yin et al., 2015). As the episode developed, the mean density shifted to lower values, indicating the
increasing contribution of less-massive carbonaceous materials. The averaged density distribution was
broadened as the episode developed, suggesting that it could be deconvolved into two groups and that the
number fraction of the low-density group increased. This finding revealed that the less-massive particles
are less hygroscopic whereas the larger density group corresponds to the more-hygroscopic one. In



addition, the variations of hygroscopicity and effective density coincided with the evolution of PBL height,
indicating that the increasing accumulation of local emissions due to adverse atmospheric conditions is
likely responsible for the enhancement of those near-hydrophobic and less-massive particles.
Figure 7 displays the temporal profiles for contributions of EC (including bare EC and OC-coated EC),
OC, sulfate, and nitrate determined by SPAMS. Obviously, the relative contribution of nitrate increased
as the episode developed. In contrast, the relative contribution of sulfate displayed an opposite trend. This
feature is comparable with the aforementioned results of SNA, thus further highlighting the important
role of nitrate in haze formation in Shanghai. The number fraction of EC particles generally increased
during the haze period, peaking at midnight on January 9 and 10. It should be pointed out that the
measured number fraction possibly underestimated the contribution of EC particles because the dominant
size range of fresh traffic particles is below the detection limit of SPAMS (0.2–2.0 μm). This finding
provides good support for the increase of near-hydrophobic and less-dense particles as the episode
developed. Niu et al. (2016) reported that the number ratio of secondary particles to soot in haze samples
was higher than that collected in the clean days in Beijing. Our finding is comparable to their results. In
contrast, the number fraction of pure OC decreased during the pollution event. The possible explanation
is that the condensation of organic matter was favored on the large amount of preexisting EC particles, or
that photo-oxidation of VOCs was minimized due to lower solar radiation.
**3.6 Evolutions of hygroscopicity and effective density with particle growth**

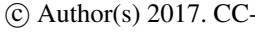



324 Three "banana-shape" evolutions of the particle size distribution were identified in the representative

325 PM episode. The banana-type contour plot of particle size distributions is a typical characteristics of new

326 particle formation (NPF) events and traditionally regarded as one of the most important criteria for

327 identifying NPF (Xiao et al., 2015;Dal Maso et al., 2005;Levy et al., 2013;Zhang et al., 2012).

328 Atmospheric NPF is often defined by the burst of nucleation mode particles and subsequent growth of the

329 nuclei to larger particles (Zhang et al., 2012;Kulmala et al., 2012). Gas-phase sulfuric acid produced via

330 oxidation of $SO_2$ by OH radical plays a dominant role in the NPF events. NPF is typically completely

331 suppressed when preexisting particles is abundant, because gas-phase sulfuric acid is rapidly lost to the

332 surfaces of preexisting aerosols (Zhang et al., 2012). In addition to sulfuric acid, low-volatility organic

333 species, and interaction between sulfate and organics are important for NPF (Zhang et al., 2004;Zhao et

334 al., 2009). However, the possibility of NPF can be ignored in this study due to the absence of the burst of

335 nucleation mode particles and the high concentration of $PM_{1.0}$. The burst of Aitken mode particles was

336 attributable to rapid accumulation of traffic emissions during rush hours under stagnant atmospheric

337 conditions. The "banana-shape" evolutions were primarily caused by coagulation and condensation

338 growth, which provided an excellent opportunity to reveal the chemical mechanism of particle growth

339 during the PM episode.

340 The first "banana-shape" evolution of the particle size distribution occurred from approximately 05:00

341 to 15:00 on January 9, with increase of the particle number concentration ($N_{total}$) from $1.7 \times 10^4$ to $3.4 \times 10^4$





cm$^{-3}$ followed by a decrease trend until 17:00 (Period 1). The second "banana-shape" evolution occurred
from approximately 18:00 on January 9 to approximately 12:00 on January 10 (Period 2). The $N_{total}$
increased from $2.1 \times 10^4$ to $4.2 \times 10^4$ cm$^{-3}$ within 3 h, followed by gradual decrease of $N_{total}$ in contrast to
continuous increase of the particle mass concentration. During the growth process, the mode diameter of
the particle population increased from below 40 nm to approximately 200 nm. The third "banana-shape"
evolution began in the evening rush hours on January 10, with continuous increase of PM mass
concentration for 12 h (Period 3).
Figure 8 illustrates the evolution of particle hygroscopicity and effective density during periods 2 and
3. During the initial stage, the measured GF and effective density distributions were both bimodal, with
a dominant peak at GF = ~1.0 and $\rho_{eff}$ = ~1.0 g cm$^{-3}$, respectively. In a previous study, we found that the
number fraction of near-hydrophobic particles varied with the traffic exhaust (Ye et al., 2013). Moreover,
laboratory studies showed that the effective density of 50 nm vehicle particles was approximately 1.0 g
cm$^{-3}$ (Olfert et al., 2007;Park et al., 2003;Momenimovahed and Olfert, 2015). These findings indicate that
the initial burst of Aitken mode particles is attributable to the presence of enhanced traffic-related
emissions. In contrast, the number fraction and GF of the more-hygroscopic group increased with growing
particle size, indicating the addition of hygroscopic inorganic species. The variation of the effective
density of the particles was similar to that of the hygroscopicity, indicating the increase of high density
materials. In general, inorganic sulfate and nitrate are more hygroscopic and denser than soot particles or



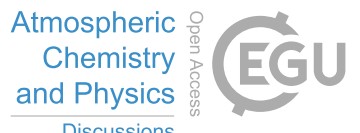

organic aerosols (Yin et al., 2015). These findings suggest that secondary sulfate and nitrate increased
with the growing particle size, indicating the importance of the conversion of $SO_2$ and $NO_x$ in particle
growth. This conclusion is supported by the largest SNA concentration in $PM_{1.0}$ during the PM episode
(31.3 μg m$^{-3}$ on January 10 and 23.8 μg m$^{-3}$ on January 11). Considering that the concentration of nitrate
was much higher than that of sulfate during the haze event, the increase of hygroscopicity was dominated
by the addition of nitrate.

**4. Conclusions**
Particle size distribution, size-resolved hygroscopic growth and effective density of sub-micrometer
aerosols were determined using a HTDMA-APM system along with measurements of cascade samples
and single particle mass spectrometry in urban Shanghai during winter 2014.
The PM episode exhibited a periodic cycle of ~5 days. The average concentration of $PM_{2.5}$ was
87.2±67.2 μg m$^{-3}$, with approximately 62% of hourly $PM_{2.5}$ concentrations exceeding the Chinese Grade
II guideline. Both secondary inorganic salts and carbonaceous aerosols contributed substantially to haze
formation, because the SNA/$PM_{1.0}$ ratio was almost constant during the observation period. Nitrate
became the most abundant ionic species at $PM_{1.0}$ >40 μg m$^{-3}$, indicating that the sources of nitrate
contributed more to haze formation in Shanghai than did $SO_2$.
The severe haze pollution was likely triggered by the adverse meteorological conditions, which caused



a large accumulation of local emissions and subsequent rapid growth to larger particles. As the PM
episode developed, the number fraction of near-hydrophobic particles of different size increased,
consistent with decrease of the mean effective density. Both hygroscopicity and effective density of the
particles were found to increase considerably with growing particle size, indicating that secondary aerosol
formation was a major contributor to particle growth. Our results suggest that the accumulation of local
emissions under adverse meteorological conditions and subsequent rapid particle growth by secondary
processes are primarily responsible for the haze pollution in Shanghai during wintertime.

**Acknowledgments**
This work was supported by the National Natural Science Foundation of China (21477020, 21527814,
and 91544224), and the National Science and Technology Support Program of China (2014BAC22B01).

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



**Figure and Table Captions**


Figure 1 Schematic diagram of HTDMA-APM system.
Figure 2. Temporal evolutions of $PM_{1.0}$, $PM_{2.5}$, and $PM_{10}$ concentrations during the winter observation.
Figure 3 Variations of sulfate, nitrate, and ammonium concentrations as a function of $PM_{1.0}$ mass loading.
Figure 4 Box plots showing hygroscopicity parameter and effective density at each dry diameter over the
whole observation. The whiskers represent the $5^{th}$ and $95^{th}$ percentile, the two borders of box display the
$25^{th}$ and $75^{th}$ percentile, and the band in each box denotes the median.
Figure 5 Temporal evolutions of particle number size distribution (A), volume size distribution (B), total
number concentration and total volume concentration (C), and $PM_{1.0}$ concentration and calculated PM
(less than 600 nm in mobility diameter) concentration during the representative PM episode from 7 to 12
January.
Figure 6 Evolutions of particle hygroscopic growth factor and effective density for different sizes during
the representative PM episode.
Figure 7 Temporal evolutions of chemical compositions determined by SPAMS during the representative
PM episode.
Figure 8 Particle hygroscopicity and density during the two particle growth processes







Figure 1 Schematic diagram of HTDMA-APM system.






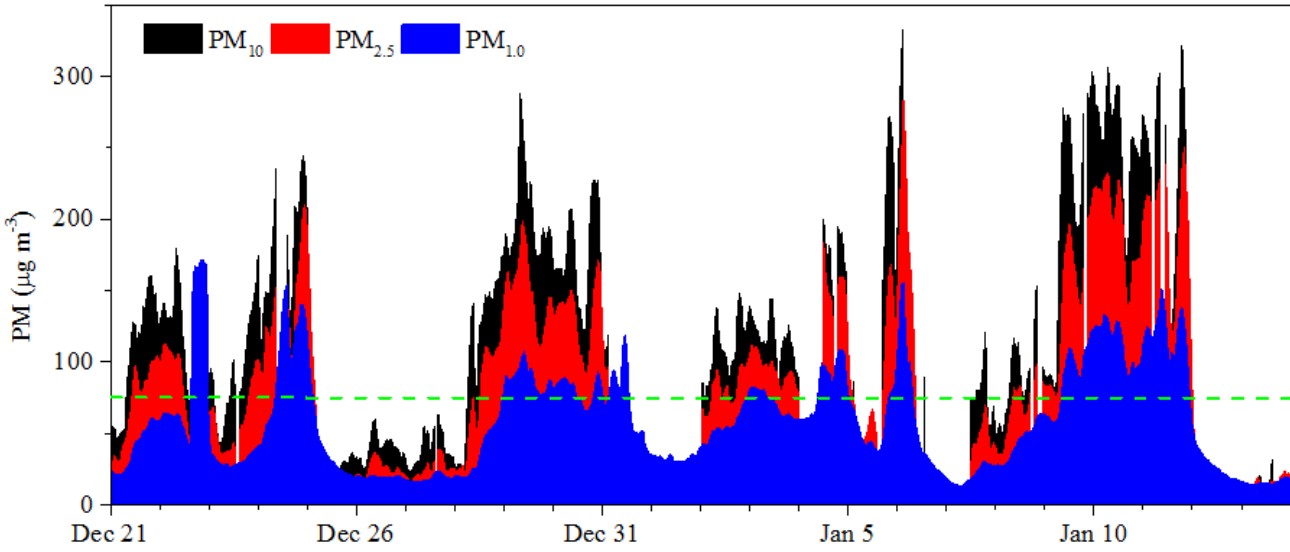


Figure 2. Temporal evolutions of $PM_{1.0}$, $PM_{2.5}$, and $PM_{10}$ concentrations during the winter observation.







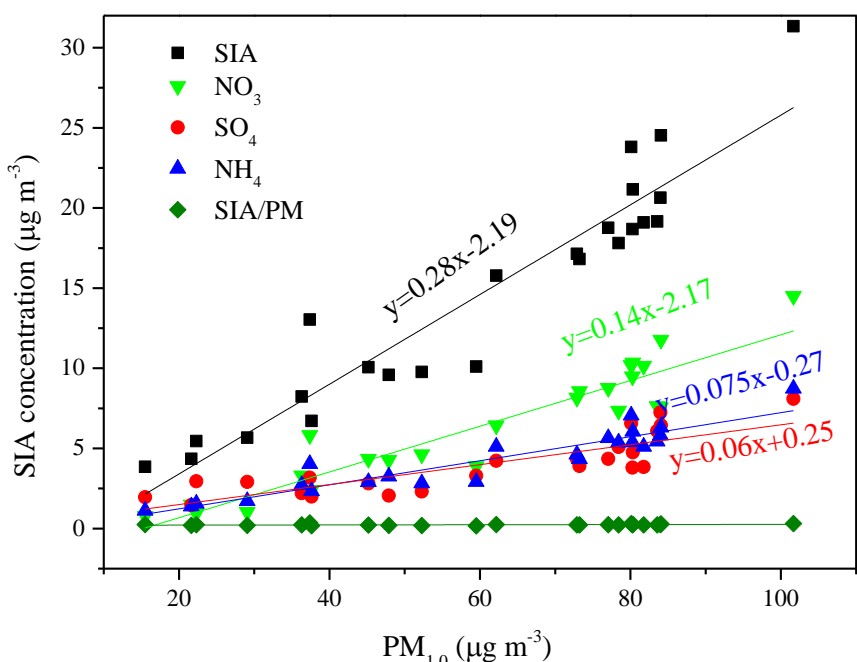


Figure 3 Variations of sulfate, nitrate, and ammonium concentrations as a function of $PM_{1.0}$ mass loading





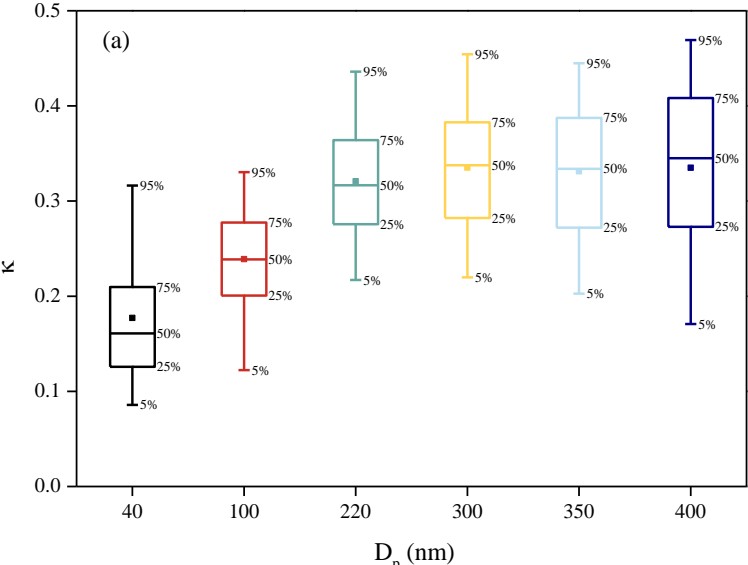


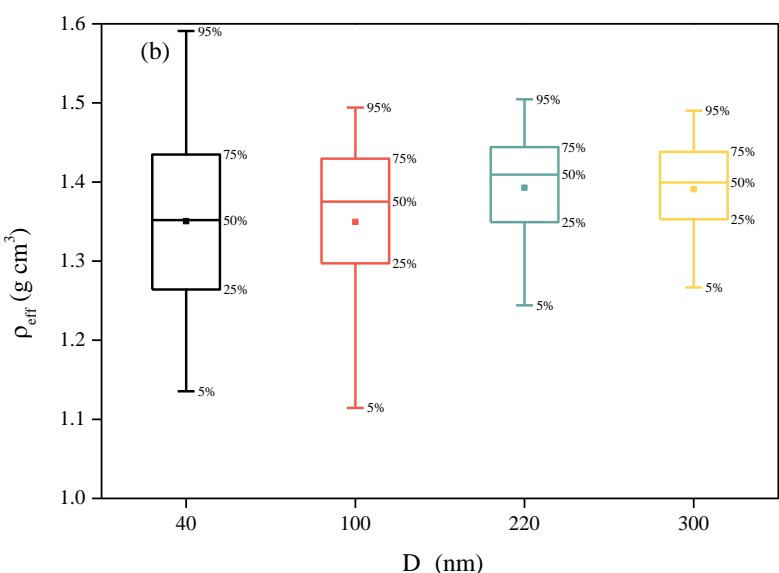


Figure 4 Box plots showing hygroscopicity parameter and effective density at each dry diameter over the
whole observation. The whiskers represent the 5th and 95th percentile, the two borders of box display the
25th and 75th percentile, and the band in each box denotes the median.






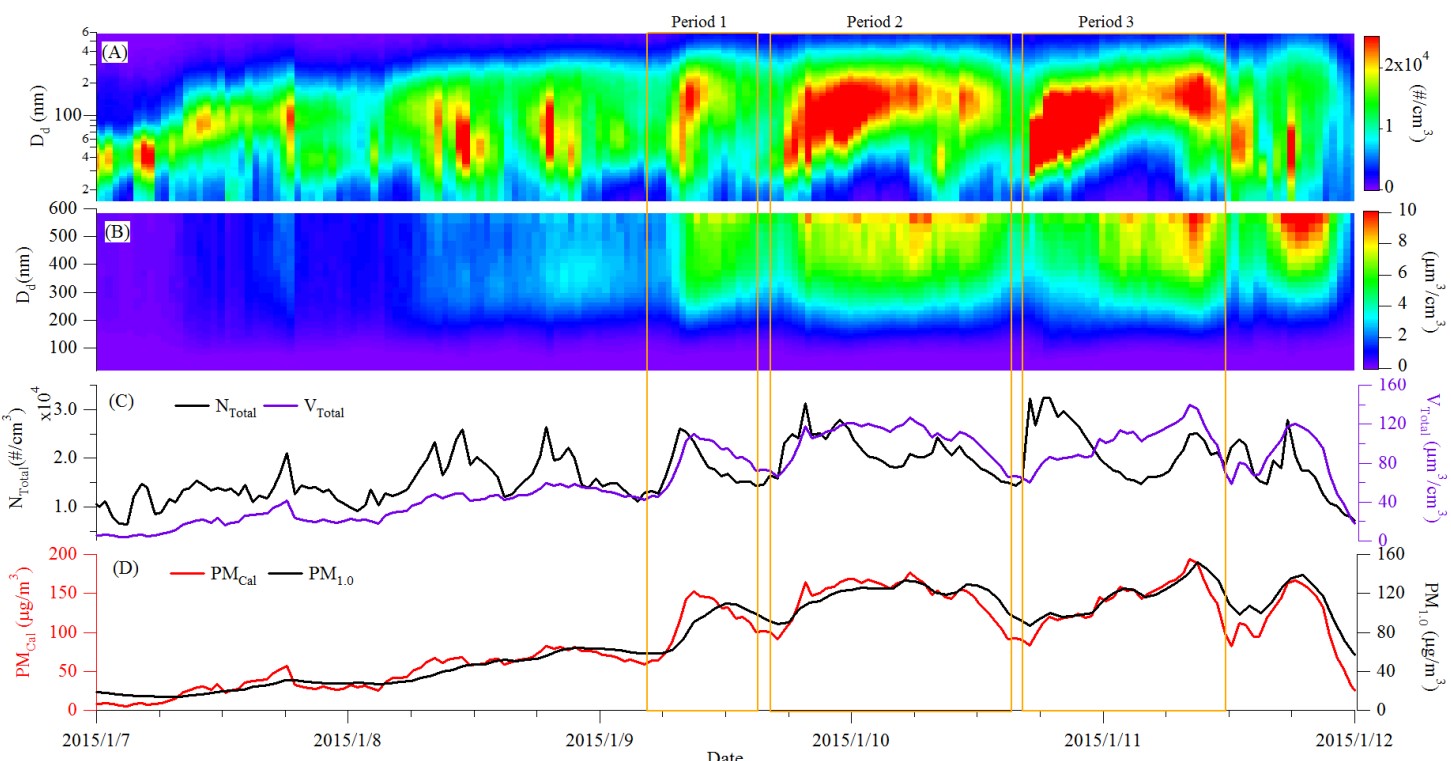


Figure 5 Temporal evolutions of particle number size distribution (A), volume size distribution (B), total
number concentration and total volume concentration (C), and $PM_{1.0}$ concentration and calculated PM
(less than 600 nm in mobility diameter) concentration during the representative PM episode from 7 to 12
January.





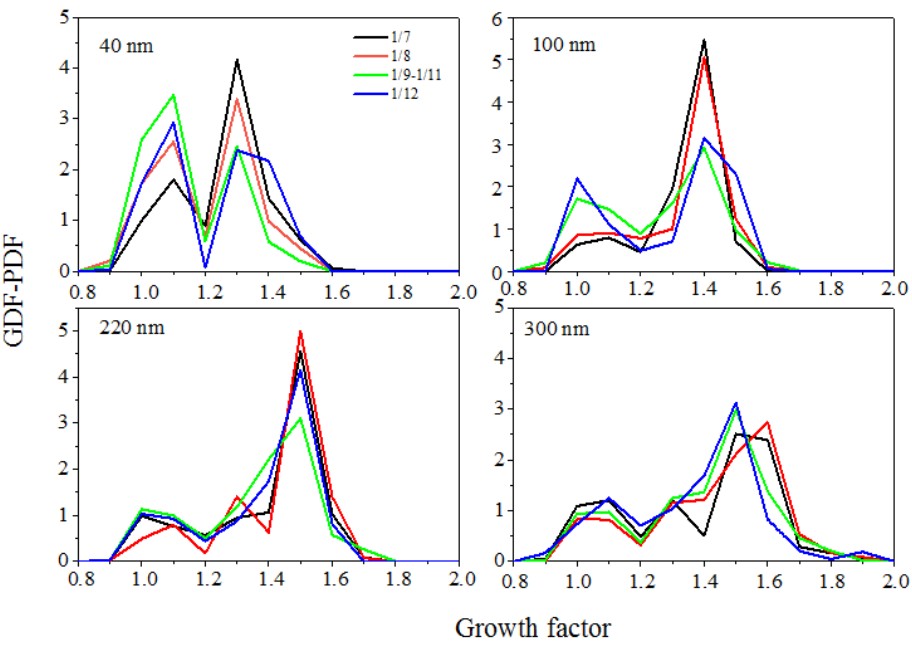


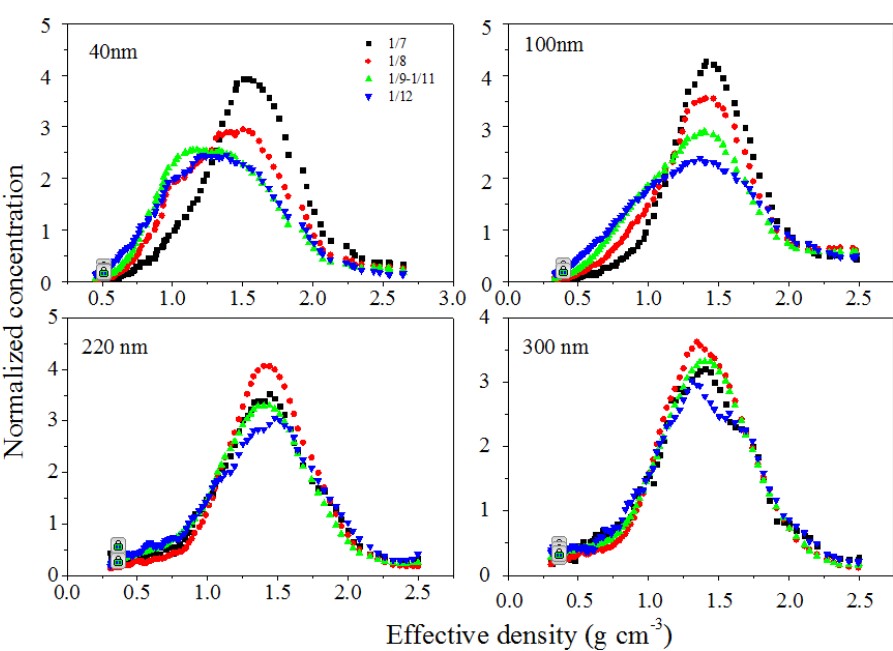


Figure 6     Evolutions of particle hygroscopic growth factor and effective density for different sizes during
the representative PM episode.








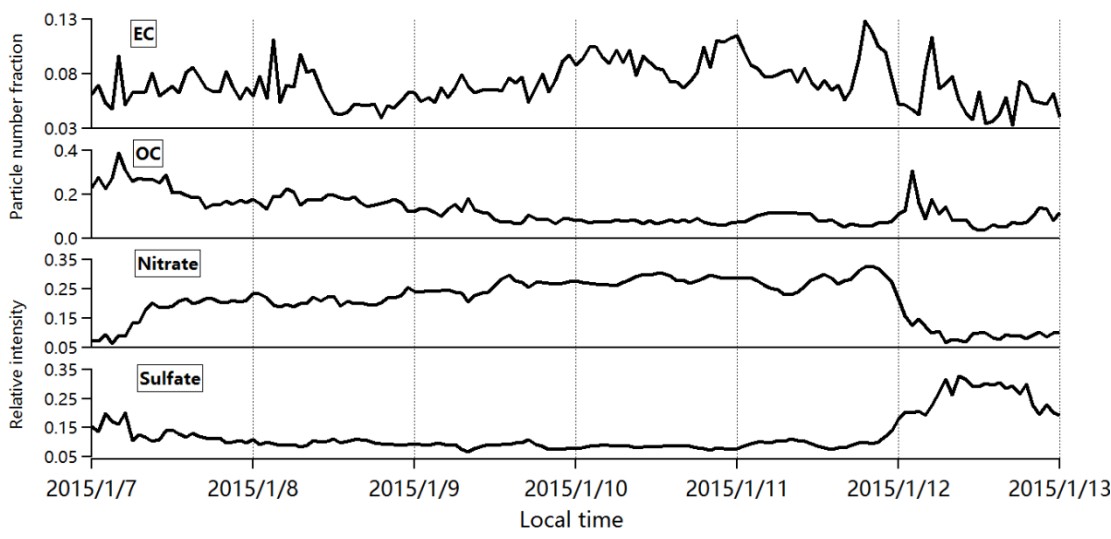


662 Figure 7 Temporal evolutions of chemical compositions determined by SPAMS during the representative

663 PM episode.






Figure 8 Particle hygroscopicity and density during the two particle growth processes.