# Peer review of "Insight into winter haze formation mechanisms based on aerosol hygroscopicity and effective density measurements"

_Atmospheric Chemistry and Physics, 2017_

## Referee Comment (RC1) · Anonymous Referee #2 · 28 Feb 2017

In this study, aerosol measurements were performed over about three weeks during winter to understand the causes of severe haze pollution in Shanghai. The measured aerosol properties include particle size distributions, hygroscopicity, effective density, and chemical composition. From the analysis of aerosols, trace gases, and meteorological data, it is concluded that the particle pollution events are caused by the accumulation of local emissions under stagnant meteorological conditions and exacerbated by rapid particle growth via secondary processes. Overall, the study is well executed, data analysis is mostly appropriate, and the paper is reasonably well written. I believe that it would be beneficial to extend the analysis to include several other effects, as detailed below. Also, a number of minor issues need to be addressed before the paper

can be accepted for publication.

A recent publication by Wang, G., et al. (Persistent sulfate formation from London Fog to Chinese haze. Proc. Natl. Acad. Sci. USA 2016, 113 (48), 13630-13635) has shown that in two other major Chinese cities the aqueous oxidation of SO2 by NO2 in the absence of light can lead to efficient sulfate formation on fine aerosols. The process requires high relative humidity and the presence of NH3. It is suggested that in heavily polluted environments, this heterogeneous process can form large amounts of particulate sulfate and nitrate in aqueous particles. Do you have photoactinic light intensity measurements to evaluate the relative contributions from photochemical and dark reactions leading to the particle growth? Were ammonia measurements available for the study period? Can you use particle hygroscopicity measurements reported in your study to derive aerosol state (aqueous/dry) and relate with the particle growth rates? Doing so would bring this study to an entirely new level.

The authors should at least attempt to explain the 5-day cycle. Was it related to the workweek/weekend cycle or something else?

Minor comments:

L11: Particulate matter (PM) and haze are not synonymous, strictly speaking. The former term is typically used to describe aqueous aerosol particles (deliquesced, but not cloud droplets). These two terms cannot be interchanged; such use creates confusion. I suggest revising the use of haze and PM in the abstract and throughout entire manuscript.

L15: This sentence may become clearer if re-written as follows: "The mass ratio of SNA/PM1.0 (sulfate, nitrate, and ammonium) fluctuated only slightly around 0.28, suggesting that both secondary inorganic compounds and carbonaceous aerosols contributed substantially to the haze formation, regardless of pollution level." Also, the original sentence implies that all of the non-SNA material is carbonaceous. Perhaps this must be stated explicitly.

L77: This statement implies that all traffic particles are soot aggregates, which is not correct

L78: Do the authors refer to material density or effective density?

L85: Must be 'cascade impactor' here and throughout the rest of the manuscript

L87: Mass spectrometry is used to measure the particle composition, which is used to infer the particle hygroscopicity and density.

L112: HTDMA does not measure the particle number size distribution

L132: '...Mass SpectrometER'

L166: these values must be rounded off, e.g., 57 +/- 37

L175: what does 'late' refer to?

L188: This sentence is confusing because it compares the contribution from a chemical (NOx) with that from a source of a chemical (presumably SO2) – coal-fired power plants. Also, doesn't coal combustion release NOx as well? The authors must provide data showing that traffic contributes more to the NOx burden than the power plants and other industrial sources that utilize coal.

L194: what does 'their' refer to?

L195: Isn't sulfate also of secondary origin?

L209: The meaning of this sentence is unclear. Why was hygroscopicity limited to smaller sizes? Do you mean 'measurements were limited to sizes smaller than 250 nm'?

L226: replace 'contradictory' with 'opposite'

L240: Insert a reference to Figure 2 early on in this paragraph

L282: Not all VOCs react with ozone. Can you provide data on the concentration of

unsaturated organics?

L286: '. . .were less- and some that were more' - what?

L304 and several other instances: 'less-massive' – did you mean 'lower density'?

L381: '. . .contributed substantially. . .because the . . .ratio was almost constant. . .' – this is an invalid argument. The second part does not follow from the first part.

Figure 2: explain in figure caption the meaning of the dashed line

Figure 3: What is 'SIA' in figure legend. Use a secondary Y-axis for the SIA/PM ratio
* * *

---

## Referee Comment (RC2) · Anonymous Referee #1 · 8 Mar 2017

The authors presented comprehensive aerosol dataset observed from metropolitan city of Shanghai. The measurements and data are valuable to study nowadays severe haze in China. The authors conclude that the accumulation of local emissions under stagnant meteorological conditions as well as rapid particle growth by secondary processes are primarily responsible for the haze formation in Shanghai. The analysis of particles hygroscopisity and density variations during pollution events is very interesting although no specific mechanism, which is actually very complex in urban areas, is addressed in the study. And also, the authors may need to improve the language. In general, I think the paper is suitable for publication in this special issue after addressing some minor issues as follows,

[Figure]

L36 remove "in heavily polluted areas" .

L88 no mechanisms are actually discussed in this paper.

Section 2.1, Besides the sampling sites information, the authors also present measurements and data information here.

L172 the authors think that the differences among the concentrations of PM1, PM2.5 and PM10 were insignificant. Is that true? According to the Fig. 2, on 26 Dec, they showed large differences in PM1, PM2.5 and PM10. L176 -177, you mentioned that the PM mass dropped sharply due to the atmospheric dilution or precipitation. Do you have such data to support this?

L182, is it 0.28, or 0?

L202, it seems particles with Dp>300 nm are with lower kappa, why? Some explanations are needed here.

L217ïijŽ The interpretation ". . .strong formation of sulfate and nitrate" looks contrary with the section 3.2, the section 3.2 shown that SNA (sulfate, nitrate, ammonium) only accounts for 28% of PM1.0.

L235-238: The reviewer is confused that why the number fraction of the lower density group increased as the concentration of NO increase. Did the authors analyze the relationship of them? or any reference?

Section 3.3, you talk about Kappa in the first part of this section, but you used GF in the second part. It'd better to use one parameter.

Fig.3 the authors may look the mass fraction of SIA, but not the mass concentrations. It is of course that the mass of each component will increase with the increase of PM.

Fig.5 and L259-261, it seems it's difficult to see the characteristics you described here. You may replot the figure to make it more clearly to reviewers.

[Figure]

L342-344, is the first banana shape a NPF event? Because you said the other two are not.

L360-371: The science of the analyzing method is weak. Anytime the number fraction and GF of the more-hygroscopic group always increase with particle size (Figure 6). It's hard to say that the hygroscopicity difference in size was caused by the particle growth. The density difference has also the similar problem.

Section 3.6, it's very interesting to look at the particles hygroscopisity and density evolutions during the particle growth. The authors have investigated the particles with different Dp (40 nm, 100 nm and 220 nm). But to my understand, it may be more reasonable to look the GF and density with same Dp during different stage. For example, how do the GF and density of 100 nm particles changed from initial stage to growth stage?

And also, according to Fig. 7, it seems, during the period 2 and period 3, the concentrations for both the Nitrate and sulfate didn't increase and remain flat trends. How can you say that the secondary sulfate and nitrate was major contributors to particle growth during haze events?

---

## Author Response (AR1)

**A point-by-point response to the reviews**

Anonymous Referee #1

Received and published: 8 March 2017

The authors presented comprehensive aerosol dataset observed from metropolitan city of Shanghai. The measurements and data are valuable to study nowadays severe haze in China. The authors conclude that the accumulation of local emissions under stagnant meteorological conditions as well as rapid particle growth by secondary processes are primarily responsible for the haze formation in Shanghai. The analysis of particles hygroscopisity and density variations during pollution events is very interesting although no specific mechanism, which is actually very complex in urban areas, is addressed in the study. And also, the authors may need to improve the language. In general, I think the paper is suitable for publication in this special issue after addressing some minor issues as follows,

**Answer**: We sincerely thanks you for your pertinent comments and valuable suggestions. The language has been polished in the revised manuscript.

L36 remove "in heavily polluted areas".

Answer: Revised.

L88 no mechanisms are actually discussed in this paper.

**Answer**: The banana-shaped particle size distribution provide a unique chance to reveal evolutions of particle hygroscopicity and effective density due to particle growth.

Section 2.1, Besides the sampling sites information, the authors also present measurements and data information here.

Answer: Revised. Measurements and data information in this section are categorized in another section.

L172 the authors think that the differences among the concentrations of PM1, PM2.5 and PM10 were insignificant. Is that true? According to the Fig. 2, on 26 Dec, they showed large differences in PM1, PM2.5 and PM10.

Answer: The statement has been revised as "Generally, the difference between the concentrations of

PM1.0 and PM2.5 during clean periods was less significant than that in haze days".

L176 -177, you mentioned that the PM mass dropped sharply due to the atmospheric dilution or precipitation. Do you have such data to support this?

**Answer**: This conclusion is support by meteorological data in the revised Figure S1. Detailed description has been added: During the end of each PM episode, the change in weather conditions played a key role in the decrease of particle concentration. As shown in Figure S1, the prevailing winds on haze days were from the northwest. The prevailing winds during two clean periods (December 25-27 and January 12-14) were northeasterly, which bring clean air mass from East China Sea. Two cold fronts from the north swept Shanghai on December 31 and January 6, bringing gale and lower temperature, which favored the dispersion of atmospheric pollutants.

L182, is it 0.28, or 0?

**Answer: it is 0.28**

L202, it seems particles with Dp>300 nm are with lower kappa, why? Some explanations are needed here.

**Answer**: For 300-400 nm particles, the average Kappa are similar (0.335 for 300 nm, 0.331 for 350 nm, and 0.335 for 400 nm), whereas the 5th percentile  $\kappa$  decreased with increasing size. Additional statement is given as "It is noticeable that the 5th percentile hygroscopicity decreased for dry diameter larger than 300 nm, likely due to the presence of the smallest dust particles (Gasparini et al., 2006)".

L217ïjŽ The interpretation ". . .strong formation of sulfate and nitrate" looks contrary with the section 3.2, the section 3.2 shown that SNA (sulfate, nitrate, ammonium) only accounts for 28% of PM1.0.

**Answer**: The formation of sulfate and nitrate is stronger compared to the USA site reported by Gasparini et al. (2006). The statement has been revised as "We attribute the different size dependencies of hygroscopicity among various measurement sites to the total emissions of  $SO_2$  and  $NO_x$ , which were responsible for the formation of hygroscopic sulfate and nitrate".

L235-238: The reviewer is confused that why the number fraction of the lower density group increased as the concentration of NO increase. Did the authors analyze the relationship of them? or any reference?

**Answer**: This feature was reported in two papers. The statement has been revised as: The lower density particles with  $\rho_{eff} < 1.0 \text{ g cm}^{-3}$  were attributable to fresh or partially aged traffic-related particles, because the number fraction of the lower density group in urban area was found to be consistent with the concentration of NO (indicator of traffic) (Levy et al., 2013;Rissler et al., 2014).

Section 3.3, you talk about Kappa in the first part of this section, but you used GF in the second part. It'd better to use one parameter.

Answer: The term "GF" in the second part is replaced by "Kappa" in the revised manuscript.

Fig.3 the authors may look the mass fraction of SIA, but not the mass concentrations. It is of course that the mass of each component will increase with the increase of PM.

Answer: The mass fraction of SIA is clearly reflected by linear regressions.

Fig.5 and L259-261, it seems it's difficult to see the characteristics you described here. You may replot the figure to make it more clearly to reviewers.

**Answer**: The particle number concentrations (black line in Figure 5) during haze period varied in the same range as in transition period, indicating little difference. The volume concentration (purple line) in haze days was always higher than that during transition period.

L342-344, is the first banana shape a NPF event? Because you said the other two are not.

**Answer**: The possibility of NPF can be ignored in this observation due to the absence of the burst of nucleation mode particles. Difference between NPF and the three particle growth events has been discussed in detail in the revised manuscript.

L360-371: The science of the analyzing method is weak. Anytime the number fraction and GF of the more-hygroscopic group always increase with particle size (Figure 6).

It's hard to say that the hygroscopicity difference in size was caused by the particle growth. The density difference has also the similar problem.

**Answer**: Indeed, this feature that the GF of the more-hygroscopic group increase with particle size cannot be attributed to particle growth in most cases, because the particles in different size are very likely from

different source. In this observation, particle growth process was clearly displayed by the banana-shaped evolutions of particle size distribution, which provided a unique chance for us to study hygroscopicity evolution due to particle growth.

Section 3.6, it's very interesting to look at the particles hygroscopisity and density evolutions during the particle growth. The authors have investigated the particles with different Dp (40 nm, 100 nm and 220 nm). But to my understand, it may be more reasonable to look the GF and density with same Dp during different stage. For example, how do the GF and density of 100 nm particles changed from initial stage to growth stage?

**Answer**: Temporal variation of GF for a certain size was extensively discussed in previous studies. Different from most studies, one highlight of this work is the particle growth process reflected by the "banana" shape particle size distribution. The objective is to reflect the changes in hygroscopicity and effective density as particle is growing. This statement has been added: The latter two banana-shaped evolutions lasted long enough to tracer the changes in hygroscopicity and effective density due to particle growth.

And also, according to Fig. 7, it seems, during the period 2 and period 3, the concentrations for both the Nitrate and sulfate didn't increase and remain flat trends. How can you say that the secondary sulfate and nitrate was major contributors to particle growth during haze events?

**Answer**: The relative contribution of nitrate determined by the SPAMS ( $0.2-2.0 \mu m$ ) increased visibly as the PM episode developed. Different from the total concentration in SPAMS, the HTDMA test showed that the hygroscopcity increased as the particle grew from 40 nm to 100 nm, revealing hygroscopic SNA contributed greatly to the particle growth from 40 nm to 100 nm particles.

**Reference**

Gasparini, R., Li, R. J., Collins, D. R., Ferrare, R. A., and Brackett, V. G.: Application of aerosol hygroscopicity measured at the Atmospheric Radiation Measurement Program's Southern Great Plains site to examine composition and evolution, J. Geophys. Res.-Atmos., 111, D05S12, doi:10.1029/2004JD005448, 10.1029/2004jd005448, 2006.

Levy, M. E., Zhang, R. Y., Khalizov, A. F., Zheng, J., Collins, D. R., Glen, C. R., Wang, Y., Yu, X. Y., Luke, W., Jayne, J. T., and Olaguer, E.: Measurements of submicron aerosols in Houston, Texas during the 2009 SHARP field campaign, J. Geophys. Res.-Atmos., 118, 10518-10534, 10.1002/jgrd.50785, 2013. Rissler, J., Nordin, E. Z., Eriksson, A. C., Nilsson, P. T., Frosch, M., Sporre, M. K., Wierzbicka, A.,

Svenningsson, B., Londahl, J., Messing, M. E., Sjogren, S., Hemmingsen, J. G., Loft, S., Pagels, J. H., and Swietlicki, E.: Effective density and mixing state of aerosol particles in a near-traffic urban environment, Environ Sci Technol, 48, 6300-6308, 10.1021/es5000353, 2014.

Gasparini, R., Li, R. J., Collins, D. R., Ferrare, R. A., and Brackett, V. G.: Application of aerosol hygroscopicity measured at the Atmospheric Radiation Measurement Program's Southern Great Plains site to examine composition and evolution, J. Geophys. Res.-Atmos., 111, D05S12, doi:10.1029/2004JD005448, 10.1029/2004jd005448, 2006.

Gasparini, R., Li, R. J., Collins, D. R., Ferrare, R. A., and Brackett, V. G.: Application of aerosol hygroscopicity measured at the Atmospheric Radiation Measurement Program's Southern Great Plains site to examine composition and evolution, J. Geophys. Res.-Atmos., 111, D05S12, doi:10.1029/2004JD005448, 10.1029/2004jd005448, 2006.

Gasparini, R., Li, R. J., Collins, D. R., Ferrare, R. A., and Brackett, V. G.: Application of aerosol hygroscopicity measured at the Atmospheric Radiation Measurement Program's Southern Great Plains site to examine composition and evolution, J. Geophys. Res.-Atmos., 111, D05S12, doi:10.1029/2004JD005448, 10.1029/2004jd005448, 2006.

**Anonymous Referee #2**

Received and published: 28 February 2017

In this study, aerosol measurements were performed over about three weeks during winter to understand the causes of severe haze pollution in Shanghai. The measured aerosol properties include particle size distributions, hygroscopicity, effective density, and chemical composition. From the analysis of aerosols, trace gases, and meteorological data, it is concluded that the particle pollution events are caused by the accumulation of local emissions under stagnant meteorological conditions and exacerbated by rapid particle growth via secondary processes. Overall, the study is well executed, data analysis is mostly appropriate, and the paper is reasonably well written. I believe that it would be beneficial to extend the analysis to include several other effects, as detailed below. Also, a number of minor issues need to be addressed before the paper can be accepted for publication.

A recent publication by Wang, G., et al. (Persistent sulfate formation from London Fog to Chinese haze. Proc. Natl. Acad. Sci. USA 2016, 113 (48), 13630-13635) has shown that in two other major Chinese cities the aqueous oxidation of SO2 by NO2 in the absence of light can lead to efficient sulfate formation on fine aerosols. The process requires high relative humidity and the presence of NH3. It is suggested that in heavily polluted environments, this heterogeneous process can form large amounts of particulate sulfate and nitrate in aqueous particles. Do you have photoactinic light intensity measurements to evaluate the relative contributions from photochemical and dark reactions leading to the particle growth? Were ammonia measurements available for the study period? Can you use particle hygroscopicity measurements reported in your study to derive aerosol state (aqueous/dry) and relate with the particle growth rates? Doing so would bring this study to an entirely new level.

**Answer**: We sincerely thanks you for your pertinent comments and valuable suggestions. The publication by Wang et al (2016) provided a new insight into night formation mechanisms of  $PM_{2.5}$  and pointed out us the research direction in the future. However, the correlation between particle growth rate and aerosol water content cannot be obtained in this study, because RH-dependent hygroscopic growth was not measured in the observation.

The authors should at least attempt to explain the 5-day cycle. Was it related to the workweek/weekend cycle or something else?

**Answer**: The periodic PM episodes are really unrelated to weekend cycles. Detailed description has been added: During the end of each PM episode, the change in weather conditions played a key role in the decrease of particle concentration. As shown in Figure S1, the prevailing winds on haze days were from the northwest. The prevailing winds during two clean periods (December 25-27 and January 12-14) were northeasterly, which bring clean air mass from East China Sea. Two cold fronts from the north swept Shanghai on December 31 and January 6, bringing gale and lower temperature, which favored the dispersion of atmospheric pollutants.

Minor comments:

L11: Particulate matter (PM) and haze are not synonymous, strictly speaking. The former term is typically used to describe aqueous aerosol particles (deliquesced, but not cloud droplets). These two terms cannot be interchanged; such use creates confusion. I suggest revising the use of haze and PM in the abstract and throughout entire manuscript.

**Answer**: I agree with you that particulate matter and haze are not the same. Particulate matter (PM) are microscopic solid or liquid matter suspended in the atmosphere (https://en.wikipedia.org/wiki/Particulates). Generally, haze pollution in china is defined as visibility decrease caused by the increase of fine particulate matter. To avoid confusion, the term "haze episode" was replaced by "haze event" in the revised manuscript.

L15: This sentence may become clearer if re-written as follows: "The mass ratio of SNA/PM1.0 (sulfate, nitrate, and ammonium) fluctuated only slightly around 0.28, suggesting that both secondary inorganic compounds and carbonaceous aerosols contributed substantially to the haze formation, regardless of pollution level." Also, the original sentence implies that all of the non-SNA material is carbonaceous. Perhaps this must be stated explicitly.

Answer: This sentence has been revised following your suggestions.

L77: This statement implies that all traffic particles are soot aggregates, which is not correct **Answer**: The nascent larger traffic particles are aggregates of primary particles with varying content of semi-volatile material. To avoid confusion, the sentence has been revised as "The effective density of nascent traffic particles varies from approximately 0.9 g cm-3 to below 0.4 g cm-3, decreasing with the increase of particle size, because there are more voids between primary particles in relatively larger aggregates (Momenimovahed and Olfert, 2015)."

L78: Do the authors refer to material density or effective density? **Answer**: Effective density. Revised.

L85: Must be 'cascade impactor' here and throughout the rest of the manuscript **Answer**: All of them has been revised following your suggestions.

L87: Mass spectrometry is used to measure the particle composition, which is used to infer the particle hygroscopicity and density.

**Answer**: We have not determined the particle hygroscopicity and density by method of chemical closure in this study. Information on particle composition measured in this study can provide some explanation to the variations of particle hygroscopicity and density. The statement has been revised as "cascade impactor samples were collected and temporal variations of particle composition were determined by a single particle mass spectrometry, which provided further insight into the hygroscopicity and density variations."

L112: HTDMA does not measure the particle number size distribution

**Answer**: Our HTDMA has the function of SMPS. Detail information on this HTDMA can see Ye et al., A multifunctional HTDMA system with a robust temperature control, Advances in Atmospheric Sciences, 26 (2009)1235-1240.

L132: '... Mass SpectrometER' Answer: The official name is Single Particle Aerosol Mass Spectrometry.

L166: these values must be rounded off, e.g., 57 +/- 37

**Answer**: Thanks for your suggestions, the sentence has been revised as "The average concentrations of PM1.0, PM2.5, and PM10 were 57 ± 37, 87 ± 67, and 129 ±78  $\mu$ g m-3, respectively."

L175: what does 'late' refer to?

Answer: The statement has been revised as "During the end of each episode".

L188: This sentence is confusing because it compares the contribution from a chemical (NOx) with that from a source of a chemical (presumably SO2) – coal-fired power plants. Also, doesn't coal combustion release NOx as well? The authors must provide data showing that traffic contributes more to the NOx burden than the power plants and other industrial sources that utilize coal.

**Answer**: Indeed, coal combustion release NOx, although NOx emission decreased significantly due to the full implement of flue gas deNOx in power plants. To avoid confusion, the statement has been revised as "This indicated that  $NO_x$  contributed more to haze formation in Shanghai compared to  $SO_2$ ."

L194: what does 'their' refer to?

**Answer**: The statement has been revised as "due to different atmospheric lifetimes among  $SO_2$ ,  $NO_x$ , and VOCs".

L195: Isn't sulfate also of secondary origin?

**Answer**: Sulfate is certainly of secondary origin. However, regional transport is a big source of SO2. So, sulfate is excluded from secondary transformation of local emissions.

L209: The meaning of this sentence is unclear. Why was hygroscopicity limited to smaller sizes? Do you mean 'measurements were limited to sizes smaller than 250 nm'?

**Answer**: The statements has been revised as "Generally, HTDMAs measure dry particles smaller than 300 nm due to technical limitations, and it is common that particle hygroscopicity increases with increase of particle size (Liu et al., 2014;Swietlicki et al., 2008)."

L226: replace 'contradictory' with 'opposite' **Answer**: Revised.

L240: Insert a reference to Figure 2 early on in this paragraph **Answer**: Revised as "As shown in Figure 2".

L282: Not all VOCs react with ozone. Can you provide data on the concentration of unsaturated organics? **Answer**: The concentration of unsaturated organics is not available in this studies.

L286: '... were less- and some that were more' - what?

**Answer**: The statement has been revised as "the nearly-hydrophobic particles were externally mixed with some hygroscopic particles".

L304 and several other instances: 'less-massive' - did you mean 'lower density'?

Answer: the term 'lower density' is replaced by 'less-massive' in the revised manuscript.

L381: '. . .contributed substantially. . .because the . . .ratio was almost constant. . .' – this is an invalid argument. The second part does not follow from the first part.

**Answer**: The statement has been revised as "Both secondary inorganic salts and carbonaceous aerosols contributed substantially to haze formation, because the mass ratio of  $SNA/PM_{1.0}$  fluctuated slightly around 0.28 during the observation period."

Figure 2: explain in figure caption the meaning of the dashed line **Answer**: Revised.

Figure 3: What is 'SIA' in figure legend. Use a secondary Y-axis for the SIA/PM ratio **Answer**: Revised following your suggestions.

**The list of all relevant changes made in the manuscript**

The line numbers are based on the ACPD version.

Line 9: "haze event from a series of periodic" was deleted

Line 10-12: "Particle size distribution, hygroscopicity, and effective density were measured online, along with analysis of water-soluble inorganic ions and single particle mass spectrometry." was revised as "Particle size distribution, hygroscopicity, effective density and single particle mass spectrometry were determined online, along with offline analysis of water-soluble inorganic ions"

Line 12-15: "Regardless of pollution level, the mass ratio of SNA/PM1.0 (sulfate, nitrate, and ammonium) slightly fluctuated around 0.28 over the whole observation, suggesting that both secondary inorganic compounds and carbonaceous aerosols (including soot and organic matter) contributed substantially to the haze formation." was revised as "The mass ratio of SNA/PM1.0 (sulfate, nitrate, and ammonium) fluctuated slightly around 0.28, suggesting that both secondary inorganic compounds and carbonaceous aerosols contributed substantially to the haze formation, regardless of pollution level"

Line 16: "During the representative PM episode," was added.

Line 16-18: "The calculated PM concentration from particle size distribution displayed a variation pattern similar to that of measured  $PM_{1.0}$  during the representative PM episode," was revised as "the calculated PM was always consistent with the measured  $PM_{1.0}$ ,"

Line 18: "the" was added between "that" and "enhanced"

Line 20: "the" was added after "indicating"; "banana-shape" was revised as "banana-shaped"

Line 21: "in  $PM_{1.0}$ " was revised as "of  $PM_{1.0}$ "; we added "that the" and "the" before "rapid size growth" and "condensation" respectively.

Line 24: "the" was added after "that"

Line 24-25: "NOx and SO2 was a major contributor to the particle growth" was revised as "NOx and SO2 was one of the most important contributors to the particle growth"

Line 32: "as well as strong impacts" was revised as "Also, atmospheric aerosol has strong impacts"

Line 33-34: "in heavily polluted areas" was deleted.

Line 63: "haze episodes" was revised as "haze events"

Line 64: "in contrast that primary emissions" was revised as "in contrast with the fact that primary emissions"

Line 65: "haze episodes" was revised as "haze events"

Line 81: "haze episodes" was revised as "haze events"

Line 82: "in the high level of sulfate during haze episodes" was revised as "about the high level of sulfate during haze events"; "It is revealed by" was revised as "It was revealed by"; "the" was added after "that"

Line 87: "which was" was revised as "as"

Line 96: "the" was added before "temporal"

Line 99: "hygroscopicity have helped the explanation of haze formation mechanisms in Beijing and Shanghai" was revised as "hygroscopicity has thrown some new light on haze formation mechanisms in Beijing and Shanghai"

Line 103-104: "effective densities of traffic particles are below 1.0 g cm-3, and density decreases with the increase of particle size because there are more voids between primary particles in relatively larger aggregates" was revised as "effective densities of nascent traffic particles varies from approximately 0.9 g cm-3 to below 0.4 g cm-3, decreasing with the increase of particle size, because there are more voids between primary particles in relatively larger aggregates"

Line 105: "density" was revised as "effective density"

Line 111-112: "cascade samples were collected and a single particle mass spectrometry was used to better understand the hygroscopicity and density variations" was revised as "cascade impactor samples were

collected and temporal variations of particle composition were determined by a single particle mass spectrometry, which provided further insight into the hygroscopicity and density variations"

Line 114: "haze episodes" was revised as "haze events"

Line 118: "The measurements" was revised as "The measurements of particle hygroscopicity and effective density"

Line 119-120: "a representative urban site close to a sub-center of Shanghai" was revised as "It can be considered as a representative urban site for Shanghai".

Line 120: "There are many dwelling quarters and commercial blocks in surrounding area. About 400 m away from the measurement site, there is the Middle Ring Line, one of the busiest elevated roads in the city." was added.

Line 120-127: A new section 2.2 "Measurements of air quality index and ground meteorological parameters" was added. "At a supersite about 100 m away" was revised as "At a supersite about 100 m away from the Environmental Building,"; "The concentrations of PM2.5, PM10, and CO" was revised as "The datas of PM2.5, PM10, and CO"

Line 128: "2.2. HTDMA-APM system" was revised as "2.3. HTDMA-APM system"

Line 154: "2.3. SPAMS" was revised as "2.4. SPAMS"

Line 165: "2.4. Ion chromatography" was revised as "2.5. Ion chromatography"

Line 166 and line 169: "Cascade aerosol samples" and "cascade samples" were revised as "Cascade impactor aerosol samples" and "cascade impactor samples"

Line 183: "PM1.0, PM2.5, and PM10 were 57.3 $\pm$ 37.0, 87.2 $\pm$ 67.2, and 127.8 $\pm$ 77.7 µg m-3" was revised as "PM1.0, PM2.5, and PM10 were 57 $\pm$ 37, 87 $\pm$ 67, and 129 $\pm$ 78 µg m-3"

Line 187-188: "During the clean period, the differences among the concentrations of  $PM_{1.0}$ ,  $PM_{2.5}$ , and  $PM_{10}$  were insignificant" was revised as "Generally, the difference between the concentrations of  $PM_{1.0}$  and  $PM_{2.5}$  during clean days was less significant than that in haze periods"

Line 189-192: "During the late episodes, the PM mass loading abruptly dropped, due to change in the atmospheric dilution or wet deposition." was revised as "During the end of each PM episode, the change in weather conditions played a key role in the decrease of particle concentration. As shown in Figure S1, the prevailing winds on haze days were from the northwest. The prevailing winds during two clean periods (December 25-27 and January 12-14) were northeasterly, bringing clean air mass from East China Sea. Two cold fronts from the north swept Shanghai on December 31 and January 6, bringing gale and lower temperature which favored the dispersion of atmospheric pollutants"

Line 200: "episodes" was replaced with "events"

Line 201-202: "This indicated that  $NO_x$  contributed more to haze formation in Shanghai than did coalfired sources" was revised as "This finding indicates that  $NO_x$  contributed more to haze formation in Shanghai compared to  $SO_2$ ."

Line 204: "haze episodes" was revised as "haze events"

Line 207: "the" was added after "with"

Line 208-209: "This finding suggests that the haze formation mechanism is likely different in Shanghai and Beijing." was revised as "This finding suggests that the haze formation mechanism in Shanghai is likely different from that in Beijing"

Line 211: "due to their different atmospheric lifetimes" was revised as "due to different atmospheric lifetimes among  $SO_2$ ,  $NO_x$ , and VOCs"

Line 215: "median hygroscopicity" was revised as "mean hygroscopicity"

Line 217: "with an average  $\kappa$  of" was revised as "with a mean  $\kappa$  of"; "the"

Line 224: "Generally, the HTDMA-measured hygroscopicity was limited to the size range below 250 nm, and it is common that the GF increases with increase of particle size." was revised as "Generally, HTDMAs measure dry particles smaller than 300 nm due to technical limitations, and it is common that particle hygroscopicity increases with the increase of particle size (Liu et al., 2014;Swietlicki et al., 2008).".

Line 225: "aerosol hygroscopicity" was replaced with "particle hygroscopicity"

Line 227: "The very few measurements for dry particles larger than 300 nm showed different size dependencies." was added into this part; "the GF first increase" was revised as "particle hygroscopicity"

Line 228: "In contrast, Wu et al. (2016c)reported that particle hygroscopicity increased with particle diameter in the range of 35-350 nm." was added.

Line 229: "no decrease in GF was observed" was revised as "the mean  $\kappa$ s of 300, 350 and 400 nm particles were nearly equal."

Line 229-230: "We attribute the different hygroscopicity to the large emissions of  $SO_2$  and  $NO_x$  in China, which were responsible for the strong formation of sulfate and nitrate." was revised as "We attribute the different size dependencies of hygroscopicity among various measurement site to the total emissions of  $SO_2$  and  $NO_x$ , gas precursors of hygroscopic sulfate and nitrate."

Line 230: "It is noticeable that the 5th percentile hygroscopicity decreased for dry diameter larger than 300 nm, likely due to the presence of the smallest dust particles." was added into this part.

Line 231: "variation" was replaced with "variability"

Line 233: "indicated" was replaced with "indicates"

Line 236: "The size distribution of particle density varied in the literature." was revised as "The size dependency of particle effective density varied in the literature."

Line 237: "particle density" was revised as "effective density" and "contradictory" was replaced with "opposite"

Line 238-239: "The difference was attributable to the contribution of fresh traffic particles" was revised as "The different trends were attributable to the variable fraction of lower density mode particles ( $\rho_{eff} < 1.0 \text{ g cm}^{-3}$ )"

Line 239: "The densities of the secondarily produced (NH4)2SO4, NH4HSO4, and NH4NO3 are ~1.75 g

cm-3. The effective density of organic aerosols varies mostly in the range of 1.2-1.6 g cm-3, depending on their source origins (Malloy et al., 2009;Turpin and Lim, 2001;Dinar et al., 2006). The lower density particles with  $\rho_{eff} < 1.0$  g cm-3 were attributable to fresh or partially aged traffic-related particles, because the number fraction of the lower density group in urban area was found to be consistent with the concentration of NO (indicator of traffic) (Levy et al., 2013;Rissler et al., 2014)." was added.

Line 240-241: "emissions from traffic exhaust" was replaced with "traffic emissions"

Line 242-243: "reported that a quasi-monodisperse density distribution was dominant for accumulation mode particles" was revised as "reported that effective density distributions were dominated by a single peak in the previous observation"

Line 243-247: "externally mixed aerosols with a lower density group ( $\rho_{eff} = \sim 1.0 \text{ g cm}^{-3}$ ) were often present in this observation, and were responsible for the decrease of the mean effective density. The lower effective density group was attributed to fresh or slightly aged traffic-related particles, because the number fraction of the lower density group increased as the concentration of NO increased." was revised as "a lower density peak below1.0 g cm-3 was often present in this observation, decreasing the mean effective density of externally mixed aerosols."

Line 249: "As shown in Figure 2," was added.

Line 251: "clean" was replaced with "clean period"

Line 257-258: "Figure 5 displays the temporal evolution of particle size distribution in comparison with the measured  $PM_{1,0}$  concentration during the representative PM episode" was revised as "Figure 5 displays the temporal profile of particle size distribution, along with the measured  $PM_{1,0}$  concentration during the representative PM episode"

Line 260: "It is noticeable that the temporal trends in mass concentrations of  $PM_{cal}$  and  $PM_{1.0}$  are highly consistent." was added into this part.

Line 265-266: "The difference of total number concentration between transition and haze periods was insignificant," was revised as "The difference of particle number concentration between transition and haze periods was less significant,"

Line 266: "rapidly" was replaced with "considerably"

Line 271: "This indicates" was revised as "This finding indicates"

Line 287: "near- hydrophobic" was replaced with "the near-hydrophobic"

Line 302, 304 and 308: "less-massive" was replaced with "lower density"

Line 324: "As shown in Figure 5," was added; "banana-shape" was revised as "banana-shaped"

Line 335: "The burst of Aitken mode particles was" was revised as "The burst of Aitken mode particles in the current study may be"

Line 337: "banana-shape" was revised as "banana-shaped"

Line 337-338: "were primarily caused by coagulation and condensation growth" was revised as "particle growth in the time evolution of particle size distribution from the Aitken mode size range to accumulation mode size range was primarily due to coagulation and condensation growth."

Line 338: "growth" was revised as "processes"; "which provided" was revised as "This feature provided"

Line 340,342,346: "banana-shape" was revised as "banana-shaped"

Line 345: "continuous increase" was revised as "a continuous increase"

Line 347: "with continuous increase" was revised as "with the continuous increase"

Line 348: "The latter two banana-shaped evolutions lasted long enough to tracer the changes in hygroscopicity and effective density due to particle growth." was added.

Line 369: "cascade samples" was revised as "cascade impactor samples"

Line 372: " $87.2\pm67.2 \ \mu g \ m^{-3}$ " was revised as " $87\pm67 \ \mu g \ m^{-3}$ "

Line 374: "because the SNA/PM1.0 ratio was almost constant during the observation period" was revised

as "because the mass ratio of SNA/PM1.0 fluctuated slightly around 0.28 during the observation period."

Line 377-378: "which caused a large accumulation" was revised as "which favored the"

Line 382: "was a major contributor to particle growth" was revised as "one of the most important contributors to particle growth"

Line 670: Figure 2 was replaced with a modified one. The caption of PM =75  $\mu$ g m-3was added into the new figure.

Line 673-676: Figure 3 was replaced with a modified one. A y axis for  $SNA/PM_{1.0}$  was added into the new figure.

**The marked-up manuscript**

**Insight into winter haze formation mechanisms based on aerosol hygroscopicity and effective density measurements**

Yuanyuan Xie1, Xingnan Ye1.2\*, Zhen Ma1, Ye Tao1, Ruyu Wang1, Ci Zhang1, Xin Yang1.2, Jianmin

Chen $1,2^*$ , Hong Chen1

[revised manuscript text omitted]
 SO2 and NOx, gas precursors of hygroscopic sulfate and nitrate. large emissions of SO2 and NOx in China, which were responsible for the strong formation of sulfate and nitrate. It is noticeable that the 5th percentile hygroscopicity decreased for dry diameter larger than 300 nm, likely due to the presence of the smallest dust particles. The variability variation—of hygroscopicity parameter  $\kappa$  was much greater for 40 nm particles. The particle population with  $\kappa < 0.1$  was attributed to fresh traffic particles (Ye et al., 2013). The considerable percentile of  $\kappa < 0.1$  indicates that the 40 nm particle population was sometimes dominated by near-hydrophobic particles.

Figure 4b displays a box chart of median effective density for different particle sizes. The median effective density varied in the narrow range of  $\rho_{eff} = 1.35 - 1.41$  g cm-3 for 40–300 nm particle population. The size dependency distribution of particle effective density varied in the literature. Hu et al. (2012) and Yin et al. (2015) reported that effectiveparticle density increased as particle size increased while a oppositecontradictory trend was observed by Geller et al. (2006) and Spencer et al. (2007). The different trends were difference was attributable to the variable fraction of lower density mode particles ( $\rho_{eff} < 1.0$  g cm-3)contribution of fresh traffic particles. The densities of the secondarily produced (NH4)2SO4, NH4HSO4, and NH4NO3 are ~1.75 g cm-3. The effective density of organic aerosols varies mostly in the

range of 1.2-1.6 g cm-3, depending on their source origins (Malloy et al., 2009;Turpin and Lim, 2001;Dinar et al., 2006). The lower density particles with  $\rho_{eff} < 1.0 \text{ g cm}^{-3}$  were attributable to fresh or partially aged traffic-related particles, because the number fraction of the lower density group in urban area was found to be consistent with the concentration of NO (indicator of traffic) (Levy et al., 2013; Rissler et al., 2014). Although the dominant accumulation mode particles have an effective density greater than Aitken mode ones, the presence of a lower effective density group associated with emissions from traffic emissions exhaust might decrease the mean effective density to a value lower than that of Aitken mode particles (Levy et al., 2014). Yin et al. (2015) reported that effective density distributions were dominated by a single peak in the previous observationa quasi-monodisperse density distribution was dominant for accumulation mode particles. In contrast, a lower density peak below1.0 g cm-3 was often present in this observation, decreasing the mean effective density of externally mixed aerosols.externally mixed aerosols with a lower density group ( $\rho_{eff} = \sim 1.0 \text{ g cm}^{-3}$ ) were often present in this observation, and were responsible for the decrease of the mean effective density. The lower effective density group was attributed to fresh or slightly aged traffic-related particles, because the number fraction of the lower density group increased as the concentration of NO increased.

**3.4 Characteristics of a representative PM episode**

As shown in Figure 2, Tthe PM episode from January 7 to 12 was a representative case of severe haze formation and elimination processes. It can be divided into clean (January 7), transition (January 8), haze

(January 9–11), and post-haze (January 12) periods. During the transition from the clean to haze period (January 7 to 8), both PM1.0 and PM2.5 concentrations increased slightly, with an average PM1.0/PM2.5 ratio of 0.65. A sharp increase in PM2.5 (of 125  $\mu$ 
[revised manuscript text omitted]

Figure 1 Schematic diagram of HTDMA-APM system.